# Co-Infection of Tomato Brown Rugose Fruit Virus and Pepino Mosaic Virus in Grocery Tomatoes in South Florida: Prevalence and Genomic Diversity

**DOI:** 10.3390/v15122305

**Published:** 2023-11-24

**Authors:** Salih Yilmaz, Ozgur Batuman

**Affiliations:** Southwest Florida Research and Education Center, Department of Plant Pathology, University of Florida, Immokalee, FL 34142, USA; salihyilmaz@ufl.edu

**Keywords:** resistance-breaking virus, inoculum, detection, new primer pairs

## Abstract

Tomato brown rugose fruit virus (ToBRFV) is an economically important seed and mechanically transmitted pathogen of significant importance to tomato production around the globe. Synergistic interaction with pepino mosaic virus (PepMV), another seed and mechanically transmitted virus, and long-distance dissemination of these two viruses via contaminated tomato fruits through global marketing were previously suggested. In 2019, we detected both viruses in several grocery store-purchased tomatoes in South Florida, USA. In this study, to identify potential sources of inoculum, co-infection status, prevalence, and genomic diversity of these viruses, we surveyed symptomatic and asymptomatic imported tomatoes sold in ten different groceries in four cities in South Florida. According to the product labels, all collected tomatoes originated from Canada, Mexico, or repacking houses in the United States. With high prevalence levels, 86.5% of the collected samples were infected with ToBRFV, 90% with PepMV alone, and 73% were mixed-infected. The phylogenetic study showed no significant correlations between ToBRFV genomic diversity and the tomato label origin. Phylogenetic analysis of PepMV isolates revealed the prevalence of the PepMV strains, Chilean (CH2) and recombinant (US2). The results of this study highlight the continual presence of PepMV and ToBRFV in imported tomatoes in Florida grocery stores.

## 1. Introduction

The viruses belonging to the Tobamovirus genus are extremely stable and mechanically transmitted plant viruses and cause significant economic losses to vegetable and ornamental crops around the globe, particularly tomatoes. In the last 60 years, diseases caused by tobamoviruses in tomatoes, such as tobacco mosaic virus (TMV) and tomato mosaic virus (ToMV), were effectively controlled by using resistance tomato cultivars harboring the *R* resistance genes [1,2,3]. The *Tm-2*^2^ and other tobamovirus resistance genes (*Tm-1* and *Tm-2*) were used to protect tomatoes against tobamoviruses until the emergence of tomato brown rugose fruit virus (ToBRFV), a recently described new tobamovirus species able to overcome these resistance genes. The pathogen was first detected in 2014 in the Middle East [2,4]. Shortly after, the spread of ToBRFV expanded globally, and the disease emerged in tomato production areas in Europe, North and South America, and Asia [5,6,7,8,9,10]. In addition to resistance breakage, synergistic interaction between ToBRFV and the potexvirus pepino mosaic virus (PepMV) was previously demonstrated, resulting in an increase in the severity of the symptoms by enhancement of PepMV titers in mixed-infected plants [11,12]. Similar to ToBRFV, PepMV is a mechanically transmitted, economically important virus, and its global epidemic has long been established since the first report on pepino (*Solanum muricatum* L.) in 1980 in Peru [13,14,15,16,17,18,19,20].

The genome of both ToBRFV and PepMV consists of a single-stranded positive-sense RNA of ~6.4 kb encapsulated in ~300 nm long rod-like particles and 508 nm long filamentous particles, respectively [2,13]. The ToBRFV virus contains four open reading frames (ORF) encoding four viral proteins, including two subunits of RNA-dependent RNA polymerase (RdRp) complex, a movement protein (MP), and a coat protein (CP) [2]. The RNA genome of PepMV consists of five ORFs encoding the RdRp, a triple gene block, and a CP [21].

ToBRFV, PepMV, and co-infection of both viruses can induce a variety of symptoms in tomato plants, and the symptom development can be affected by many factors, including environmental conditions, virus accumulation, tomato cultivar, and growth stage of the infected plants [12,22]. Generally, plants infected with ToBRFV exhibit common tobamovirus symptoms that are stunted, with leaves showing mosaic or mottling and fruit bleaching [2,4]. Characteristics of PepMV-induced symptoms are dwarfing, leaf distortions, mosaics and narrowing of the leaves, fruit discoloration, and the appearance of open fruit and scars on the fruit surface [11,23,24]. ToBRFV and PepMV co-infections in tomatoes can induce severe viral disease symptoms, including scarred or open unripe fruits and narrow or yellow patched leaves [11].

Currently, PepMV is classified into six strains/genotypes, including American (US1), Chilean 2 (CH2), the recombinant (US2) US1/CH1, European (EU), Peruvian (LP), and new Peruvian (PES) [25]. After the first identification in 2001, all four major genotypes (EU, US1, US2, and CH2) of PepMV were characterized in tomato plants in Arizona, California, Colorado, Maryland, New York, Ohio, Oklahoma, and Texas in the U.S. [26,27].

Previous phylogenetic studies of ToBRFV populations have revealed three major clades, with minor divergence among these clades [28,29,30,31]. These findings collectively highlighted the genetic diversity and potential origins of ToBRFV in various geographic regions. The detection of ToBRFV in North America was first reported at a California greenhouse facility and in Mexico in September 2018 [5,7]. Shortly after, the pathogen was reported in Canada in 2019, and greenhouse tomato production sites in other states in the U.S., including Arizona, Florida, and New Jersey [7,32]. In Florida, ToBRFV was detected in tomatoes grown in the community garden and in tomatoes imported from Mexico for sale in grocery stores around the same time in 2019 [33,34].

In the United States, California and Florida are the top two fresh tomato producers, accounting for nearly 75% of total fresh tomato production in the U.S. in the last two decades [35]. However, about 60% of domestic demand for fresh market tomatoes is supported by imported tomato fruits, mainly from Mexico and Canada. In 2020, imports from Mexico accounted for 91% of the total volume of U.S. fresh tomato imports. Canada was a distant second, accounting for 8% of U.S. imports [36]. From 2000 to 2020, the volume of tomato imports from Mexico increased nearly threefold, whereas the total U.S. production plunged 67% within the same timeframe. The imports from Mexico were three times higher than the total U.S. production in 2020 [36].

After the first disease report associated with ToBRFV in a California greenhouse in 2018, the U.S. Department of Agriculture (USDA) Animal and Plant Health Inspection Service (APHIS) issued a federal order to control tomato fruit imports from countries with ToBRFV [37]. Based on the order, fruit entering the United States from key trading partners such as Canada and Mexico would be visually inspected at the entry ports. All tomatoes imported from a country where ToBRFV is present must include a phytosanitary certificate or an inspection certificate documenting that the shipment is free of symptoms of ToBRFV. However, due to the challenging and unreliable visual symptom determination of viruses and the presence of asymptomatic fruits, infected tomato fruits may escape from visual inspection and inadvertently move across national and state borders [38].

To date in Florida, PepMV has not been reported in an open field, but recently, we detected co-infection with ToBRFV in imported greenhouse tomatoes sold in supermarkets [33]. Tomato fruits displaying symptoms consistent with ToBRFV and PepMV infections (Figure 1) were observed at several grocery stores in South Florida in the United States. Based on these observations, we hypothesized that the infectious ToBRFV and/or PepMV might still be present in the grocery tomatoes and may contribute to the disease dispersal due to their easy transmissibility and extreme durability. While the current status of open-field infection with PepMV, ToBRFV, and co-infection in Florida remains to be investigated, with no reports to date, our study has a twofold objective. First, we aim to investigate the prevalence of the ToBRFV, PepMV, and their co-infection status in imported grocery tomatoes in South Florida by broadening our investigation to encompass a wider range of grocery stores in the region. Subsequently, we aim to understand the genomic diversity and the phylogenetic relationship between the origin of the tomato produce sold in supermarkets.

## 2. Materials and Methods

### 2.1. Sample Collection and Virus Source

In 2020 and 2022, tomato fruits sold in several leading chain grocery stores were randomly inspected visually for virus-like symptoms such as necrosis and chlorosis. Suspected fruit samples with and without obvious virus-like symptoms were purchased from grocery stores located in four different cities (Arcadia, Hollywood, Immokalee, and Naples) in South Florida.

### 2.2. RNA Extraction and Virus Detection

The total RNA was extracted from symptomatic and asymptomatic fruit pericarp (~50 mg) of tomato plants using the Quick-RNA MiniPrep kit (Zymo Research, Irvine, CA, USA) according to the manufacturer’s instructions. Extracted RNAs were stored at −80 °C until needed. cDNAs were synthesized by Superscript II (200 U/μL) reverse transcriptase (Invitrogen, Carlsbad, CA, USA) using the total RNAs of virus-infected and healthy tomato fruit samples. The cDNAs were then used in RT-PCR assay and amplification initiated with ToBRFV- and PepMV-specific and Tobamo- and Potex-degenerate primers targeting RNA-dependent RNA polymerase (RdRp) or coat protein (CP) [2,7,11,39]. For ToBRFV detection, in addition to available primers, new primer pairs targeting the partial movement protein (MP) and complete CP of ToBRFV were designed and compared with others. The thermocycler program for the new primer pairs consisted of a 2 min denaturation at 94 °C, followed by 35 cycles of 94 °C for 20 s, 55 °C for 1 min, and 72 °C for 1 min, with final extension at 72 °C for 7 min. Primers used and designed in this study are summarized in Table 1.

### 2.3. Sanger Sequencing and Phylogenetic Analysis

The complete CP gene region from representative isolates of 24 ToBRFV and 28 PepMV were sent for Sanger sequencing (MCLAB, South San Francisco, CA, USA). The phylogenetic trees were constructed using a complete nucleotide sequence of CPs of PepMV and ToBRFV with other available representative isolates from South and North America, Europe, and Asia retrieved from GenBank (Table 2). All the phylogenetic trees were constructed using the Tamura-Nei (TN93) genetic distance model using Geneious Prime (v.2022.2.2) based on PHYML, RAxML, and neighbor-joining methods for ToBRFV and PHYML method for PepMV [40,41]. Bootstrapping (1000 replicates) was generated to ensure the confidence of the branches in the phylogenetic trees.

## 3. Results

### 3.1. Tomato Brown Rugose Fruit Virus (ToBRFV) and Pepino Mosaic Virus (PepMV) Detection in Grocery Tomatoes

In 2020 and 2022, a total of 124 symptomatic and asymptomatic tomato fruits under nine different brands were collected from ten different grocery stores located in four different cities (Arcadia, Hollywood, Immokalee, and Naples) in South Florida. The country of origin of the packaged tomatoes was Canada, Mexico, or unknown (i.e., repacked in the U.S.). Symptomatic tomato fruits were showing common ToBRFV and PepMV symptoms, including fruit discoloration, uneven ripening, bright yellow spots, and yellow/green “marbling” (Figure 1).

**Figure 1 viruses-15-02305-f001:**
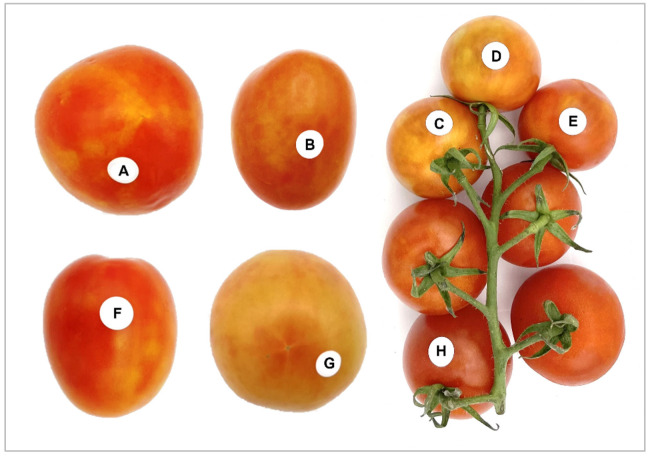
Symptoms of grocery store-purchased tomatoes in South Florida. Representative tomato fruit samples with chlorosis and blotchy symptoms were suspected to be infected with a virus. (A–G) Fruits with yellow/green “marbling” and bright yellow patches resembling common fruit symptoms caused by tobamovirus infection. (C–G) Fruit discoloration, and (H) asymptomatic fruit. (A–E,G,H) tomato fruits co-infected with tomato brown rugose fruit virus (ToBRFV) and pepino mosaic virus (PepMV), and (F) infected with ToBRFV alone. The presence of viruses was confirmed by RT-PCR. All photos courtesy of the author.

Representative 52 tomato fruits were selected and tested for the presence of ToBRFV and PepMV by RT-PCR using virus-specific and degenerate primers targeting capsid protein (CP) and/or RNA-dependent RNA polymerase (RdRp) region. Newly designed ToBRFV-specific primer pair, ToBRFV-F5281/R6308, and previously published PepMV-F5380/3END, targeting complete CP gene regions showed the highest success rate in determining the presence of viruses in 45/52 and 47/52 samples, respectively. Whereas other primers, including ToBRFV-F/R (38/52), ToBRFV-F3666/R4718 (34/52), PVX-UniF/R (33/52), and PepMV-F/R (34/52), failed to confirm the presence of the viruses in some positive tomato samples. According to the package labels, all tomato fruits tested positive for these viruses either originated from Canada and Mexico or were repacked in the United States without information on their country of origin. Samples representing tomatoes produced in Florida all tested negative and were consistent with the lack of any virus-like symptoms at the grocery stores (data not shown). The percent prevalence of viruses in these fruits showed the presence of ToBRFV, PepMV, and mixed infection with both viruses in each origin. Based on the RT-PCR results, overall, 86.5% (45/52) of the tested samples were found to be infected with ToBRFV, 90% (47/52) with PepMV, and 77% (38/52) co-infected with ToBRFV and PepMV (Table 3).

All the tested fruit samples coming from Canada were co-infected by ToBRFV and PepMV and, therefore, had the highest incidence levels (100%) of these viruses (Figure 2). Mexico had the lowest incidence levels of ToBRFV, PepMV, and co-infection with 66.7%, 80%, and 46.7%, followed by the United States at 89.5%, 89.5%, and 78.9%, respectively (Figure 2).

### 3.2. Phylogenetic Analysis of Virus Isolates from Grocery Store Tomato Fruits

To determine the genetic relationship of the PepMV population detected in the tested fruit samples, the phylogenetic comparison was made using the complete CP nucleotide sequence of PepMV isolates obtained from this study and representative isolates for each genotype from GenBank and potato virus X (PVX; MT752896) as an outgroup (Table 2). The similarity of PepMV CP sequences determined from grocery samples was found to be 96.2% to 100% identical (up to 27 single nucleotide polymorphisms, SNPs). All the PepMV isolates from this study clustered with either US2 or CH2 genotype groups (Figure 3). All the PepMV isolates coming from tested fruits of Canadian origin were closely linked to the CH2 genotype, while the majority of PepMV isolates from Mexico grouped with the US2 genotype, except for two isolates (sample ID: 22AL23 and 22AL24) that were grouped with the CH2 (Figure 3). Similar to isolates from Mexico, the phylogenetic analysis showed no uniform grouping pattern for the isolates from United States origin. However, the majority of the U.S. isolates (with unknown origin of import) grouped in the US2 genotype except for three isolates (Sample ID: 22NTT13, 22NTT14, and 20NTTG) that were linked to the CH2 group (Figure 3).

The nucleotide sequence similarity of ToBRFV CPs from this study was found to be 97.7% to 100% identical (up to 11 SNPs). To compare ToBRFV isolates, phylogenetic analyses were initially conducted by using PHYML and RAXML methods, but these results did not yield a significant distribution among the samples (results not shown). However, the phylogenetic analysis with the complete CP nucleotide sequences of ToBRFV isolates made in a neighbor-joining method in a recent study was able to show the isolate distribution into three different phylogroups [30]. Therefore, phylogenetic analysis with 24 CP nucleotide sequences of ToBRFV isolates from this study and 26 others, including tobacco mosaic virus (TMV) as an outgroup from GenBank, were made using the reported neighbor-joining method. Similar to previous phylogenetic studies, our phylogenetic analyses with this method resulted in three distinct phylogroups. The two isolates (MW284987, MW284988) from France clustered in Group 3, and all other isolates from Belgium (MZ945419, OM515261, and OM515270), the Netherlands (OM515241, MN882040, MW314130, OM515239, and MW314119), and the United Kingdom (MN182533) clustered in Group 2 (Figure 4). All the isolates from this study originated from Canada, Mexico, and the United States were clustered in Group 1. No significant correlation was observed between genomic diversity among the ToBRFV isolates and their origins.

## 4. Discussion

During this study, tomato fruits showing symptoms akin to ToBRFV and/or PepMV infections were consistently observed at various grocery stores in the cities of Arcadia, Hollywood, Immokalee, and Naples in Florida in both 2020 and 2022. To gain insights into the virus distribution, genomic diversity, and the phylogenetic relationship between the viruses and the origin of tomato produce sold in grocery stores, purchased tomato fruits were tested for ToBRFV and PepMV infections. Firstly, to validate and improve the detection of ToBRFV via RT-PCR assays, we designed new primer pairs targeting partial MP and complete CP gene sequence of the virus (Table 1). Our newly designed primer pair in this study was found to perform better in detecting ToBRFV than other widely used primers by different groups. Tomato fruits purchased from various grocery stores exhibited a high prevalence of viral infections. ToBRFV was detected in 86% of the tomato fruits, PepMV in 90%, and a significant proportion (77%) showed co-infection of both ToBRFV and PepMV. Among asymptomatic fruits, three were infected with ToBRFV, two with PepMV, and five displayed mixed infection. These results indicate that virus-infected tomato fruits with or without obvious symptoms were shipped thousands of miles from primary production areas to various consumption sites, albeit without knowing their potential to serve as an inoculum source for local production sites throughout the state. Based on the product labels, grocery store-purchased tomatoes originated from either Canada or Mexico or were redistributed from packing houses in the U.S.

Various phylogenetic analyses were conducted to better understand the genomic relationship of the ToBRFV and PepMV isolates in grocery store-purchased tomatoes and their origin of production sites. In the absence of a complete genome sequence, the phylogenetic study of tobamoviruses was often established using the complete CP sequence [42,43,44]. Our phylogenetic analysis, incorporating the complete CP gene sequence of ToBRFV isolates from this study and representative isolates from GenBank, revealed no noteworthy association between genomic diversity and geographical origins (e.g., Canada, Mexico, and the repacked U.S.). Nonetheless, subsequent phylogenetic analysis using the neighbor-joining method identified three distinct phylogroups among the ToBRFV isolates. These results were consistent with the results of the recent studies on population structure and evolutionary analysis of global ToBRFV isolates with distribution to three main clusters [28,30]. Furthermore, all the ToBRFV isolates from this study are clustered in one group (Figure 4), suggesting possible descent from a single recent common ancestor.

Previous studies have identified the presence of four major genotypes (EU, US1, US2, and CH2) of pepino mosaic virus (PepMV) in North American greenhouse tomato facilities [26]. Initially, the EU genotype was predominant in North America, contrary to the expectation of the US1 and US2 genotypes [26]. However, a significant shift occurred in 2010, when the CH2 genotype became prevalent, followed by another shift to the US1 genotype in Mexico in 2012 [27]. In our study, phylogenetic analysis indicated that the PepMV isolates from grocery store-purchased tomatoes cluster with the US2 and CH2 genotypes. The EU genotype was absent, suggesting no reverse shift to EU genotypes in North America. Similar findings were observed for PepMV isolates from Canadian tomatoes, closely linked to the CH2 genotype. Interestingly, most PepMV isolates from Mexico were grouped with the US2 genotype, except for two isolates associated with the CH2 genotype. However, the determination of how this PepMV genotype shift occurred in North America is beyond the scope of our current study. Furthermore, most PepMV isolates from the U.S., with unknown import origins, clustered within the US2 genotype, suggesting potential sources from Canada and Mexico.

Overall, our study showed the continual presence of PepMV and ToBRFV viruses in grocery store tomatoes. However, it is less likely that the distribution of such viruses in greenhouses or open fields in Florida may occur through various scenarios. Here, we present three possible scenarios to illustrate the potential spread of viruses. Scenario 1, through personnel: A greenhouse worker acquires tomatoes infected with a viral pathogen from a grocery store and consumes a sandwich prepared with these contaminated tomatoes within the working environment without knowledge of the viral infection. Consequently, the worker’s hands and clothes become contaminated. Following this, the worker gains access to the greenhouse, unintentionally spreading the viral pathogens to uninfected plants through physical contact or aerosol transmission. Scenario 2, equipment transmission: Workers handle a shipment of virus-infected tomatoes in the packing house where tomatoes are processed and sorted. Due to time constraints or inadequate cleaning practices, the equipment used in sorting, such as conveyor belts or sorting trays, is not thoroughly disinfected between batches. As a result, viral particles from the infected tomatoes remain on the surfaces of the equipment. Subsequently, if these contaminated packaging materials are reused by growers or greenhouse operators for packaging or storing their own produce, the viruses can be transferred to their facilities. Scenario 3, environmental transmission: ToBRFV viral particles originating from grocery store-purchased tomatoes can reach a community garden. These particles can be carried to a nearby greenhouse or open field through bumblebees [44] or irrigation water [45]. The viral particles can infect healthy plants in the new environment, leading to disease outbreaks. It is essential to emphasize the need for further research or experimentation to confirm these pathways and better understand pathogen transmission dynamics. Nonetheless, these scenarios are just a few examples aimed at illustrating the low-risk but possible pathways established by existing virus transmission understanding and scientific rationale through which viruses from infected grocery store-purchased tomatoes could potentially disseminate within greenhouse or open field environments in Florida.

Although currently, there are enforced pathogen screening and regulations for ToBRFV at the U.S. borders, our study suggests that these inspections (and regulations) are not effectively blocking infected tomato fruits from entering the U.S. However, recent pathway risk assessments by the United States Department of Agriculture, Animal and Plant Health Inspection Service (USDA-APHIS) suggest that imported fruits are not a primary introduction source of ToBRFV [46]. While we did detect the virus in imported tomatoes and have not found any ToBRFV infection in ongoing field surveys in Florida [33,47], our study supports this assessment and underscores how improved sanitation and disease management practices in Florida have played a vital role in decreasing the risk of secondary ToBRFV dissemination through imported produce into tomato growing areas.

In light of these findings, and considering the increased understanding of disease management globally, there is merit in more carefully evaluating the potential of virus-infected tomatoes to serve as an inoculum source for Florida production sites in the future. Such evaluation should, however, be accompanied by comprehensive training and risk awareness programs targeting stakeholders involved in the production and distribution of tomatoes. For this to happen in the near future, certain restrictions associated with ToBRFV and PepMV research in the U.S. should be revisited. Such consideration could offer researchers an opportunity to conduct more in-depth studies on these exotic viruses, facilitating a better understanding of the virus etiology and epidemiology under our growing practices and field conditions. This will contribute to the development of enhanced control measures should these viruses be introduced to Florida fields.

Ultimately, a well-informed decision-making process, characterized by collaboration among regulatory authorities, researchers, growers, and industry experts, will be instrumental in striking an appropriate balance between scientific exploration and the protection of agricultural systems. We can foster a safer and more robust food production and distribution network by adhering to this approach and understanding the critical factors involved in introducing and spreading exotic tomato-infecting viruses to production regions through imported produce.

## Figures and Tables

**Figure 2 viruses-15-02305-f002:**
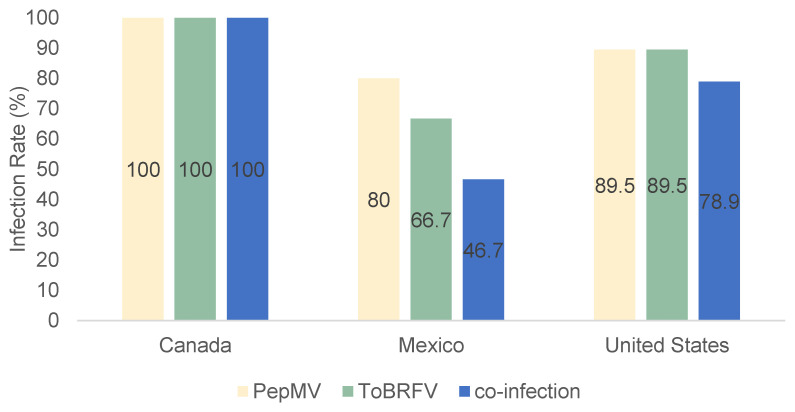
The percent distribution of tomato brown rugose fruit virus (ToBRFV) and pepino mosaic virus (PepMV) and their co-infection in tomatoes imported from Canada, Mexico, and the United States. (The origin of the tomatoes was determined based on the package labels).

**Figure 3 viruses-15-02305-f003:**
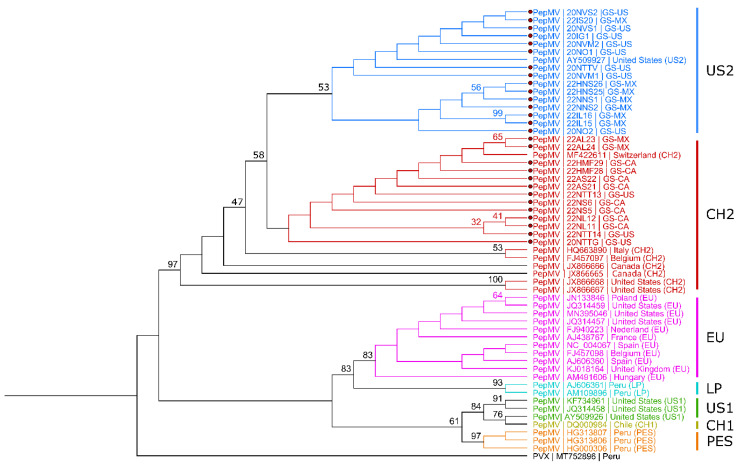
Phylogenetic analysis of pepino mosaic virus (PepMV). Phylogenetic inferences using maximum likelihood (PHYML) phylogenetic analysis using Tamura-Nei (TN93) genetic distance model with uniform rates among sites (1000 bootstrap replicates, only >30% values were shown) by Geneious Prime^®^ (2022.2.2) software to show the distribution of 27 PepMV isolates from this study (red dots) and other genotypes from GenBank representing each phylogroup based on complete CP gene nucleotide sequences. Phylogroups are presented in blue: US1; red: CH2; pink: EU; turquoise: LP; green: US1; yellow: CH1; orange; PES black; potato virus X (PVX), outgroup. Isolate names from this study ending with MX, CA, and U.S., representing fruits’ origin being from Mexico, Canada, and the U.S. (with unknown origin of import), respectively.

**Figure 4 viruses-15-02305-f004:**
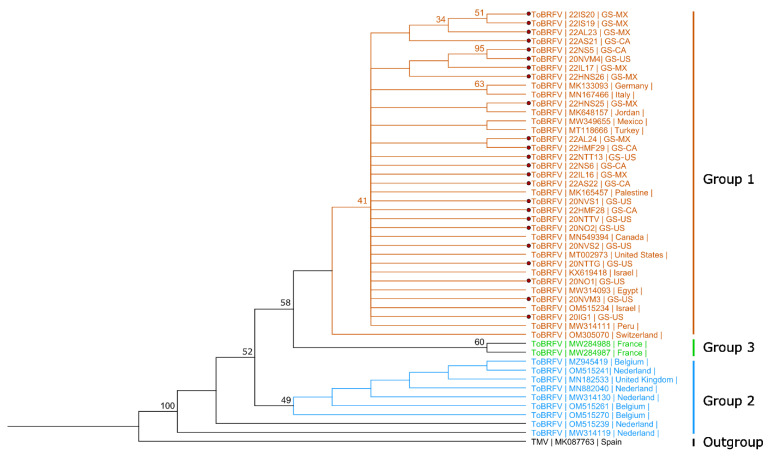
Phylogenetic analysis of tomato brown rugose fruit virus (ToBRFV) isolates. Twenty-four ToBRFV isolates from this study (red dots) and other isolates representing each phylogroup based on the complete nucleotide sequence of the CP gene from GenBank were used in the phylogenetic analysis. Neighbor-joining phylogenetic analysis using Tamura-Nei (TN93) genetic distance model with uniform rates among sites (1000 bootstrap replicates, only >30% values were shown) by Geneious Prime^®^ (2022.2.2) software was utilized. Phylogroups are presented in orange: Group 1; blue: Group 2; green: Group 3; and black: outgroup. Isolate names from this study ending with MX, CA, and U.S. represent fruits origin being from Mexico, Canada, and the U.S. (with unknown origin of import), respectively.

**Table 1 viruses-15-02305-t001:** The list of RT-PCR primers used in this study for the detection of tomato brown rugose fruit virus (ToBRFV) and pepino mosaic virus (PepMV) in tomato fruits.

Primer Name	Sequence (5′-3′)	Target	Amplicon (bp)	Reference
ToBRFV-F	GAAGTCCCGATGTCTGTAAGG	partial CP	842	[7]
ToBRFV-R	GTGCCTACGGATGTGTATGA			
ToBRFV-F5281 ToBRFV-R6308	AGGACGCAGAAAAGGCAGTT	complete CP	1028	This study
	CCATACACATTTGTCCCGCG			
F-3666 R-4718	ATGGTACGAACGGCGGCAG CAATCCTTGATGTGTTTAGCAC	partial RdRp	1052	[2]
PVX-UniF	ACNTAYGCNGGHTGYCARGG	partial RdRp	1100	[39]
PVX-UniR	CCATNGTHCCYWANAMCATNAC			
PepMVF	GAGCTGTGGATTCCATCC	partial RdRp	835	[39]
PepMVR	CAACCTTGTTTAACAAATTGG			
PepMV-F5380	CACCAATAAATTTAGTTTTAGC	complete CP	1035	[11]
PepMV-3END	ATTTAGTAGATTTAGATACTAAGG			

**Table 2 viruses-15-02305-t002:** The list of complete capsid protein (CP) gene nucleotide sequences of tomato brown rugose fruit virus (ToBRFV) and pepino mosaic virus (PepMV) isolates used in this study.

Virus (Strain)	Origin	Host	Sample ID	Accession Number
PepMV (CH2)	United States	*Solanum lycopersicum* L.	20NTTG *	OP971908
	United States	*Solanum lycopersicum* L.	22NTT13 *	OP971931
	United States	*Solanum lycopersicum* L.	22NTT14 *	OP971932
	Canada	*Solanum lycopersicum* L.	22NS5 *	OP971929
	Canada	*Solanum lycopersicum* L.	22NS6 *	OP971930
	Canada	*Solanum lycopersicum* L.	22NL11 *	OP971925
	Canada	*Solanum lycopersicum* L.	22NL12 *	OP971926
	Canada	*Solanum lycopersicum* L.	22HMF28 *	OP971918
	Canada	*Solanum lycopersicum* L.	22HMF29 *	OP971919
	Canada	*Solanum lycopersicum* L.	22AS21 *	OP971916
	Canada	*Solanum lycopersicum* L.	22AS22 *	OP971917
	Mexico	*Solanum lycopersicum* L.	22AL23 *	OP971914
	Mexico	*Solanum lycopersicum* L.	22AL24 *	OP971915
PepMV (US2)	Mexico	*Solanum lycopersicum* L.	22NNS1 *	OP971927
	Mexico	*Solanum lycopersicum* L.	22NNS2 *	OP971928
	Mexico	*Solanum lycopersicum* L.	22HNS25 *	OP971920
	Mexico	*Solanum lycopersicum* L.	22HNS26 *	OP971921
	Mexico	*Solanum lycopersicum* L.	22IL15 *	OP971922
	Mexico	*Solanum lycopersicum* L.	22IL16 *	OP971923
	Mexico	*Solanum lycopersicum* L.	22IS20 *	OP971924
	United States	*Solanum lycopersicum* L.	20NO1 *	OP971906
	United States	*Solanum lycopersicum* L.	20NO2 *	OP971907
	United States	*Solanum lycopersicum* L.	20IG1 *	OP971905
	United States	*Solanum lycopersicum* L.	20NTTV *	OP971909
	United States	*Solanum lycopersicum* L.	20NVM1 *	OP971910
	United States	*Solanum lycopersicum* L.	20NVM2 *	OP971911
	United States	*Solanum lycopersicum* L.	20NVS1 *	OP971912
	United States	*Solanum lycopersicum* L.	20NVS2 *	OP971913
PepMV (LP)	Peru	*Solanum peruvianum* L.	NA	AJ606361
	Peru	*Solanum muricatum* L.	NA	AM109896
PepMV (PES)	Peru	*Solanum lycopersicum* L.	NA	HG000306
	Peru	*Solanum peruvianum* L.	NA	HG313807
	Peru	*Solanum lycopersicum* L.	NA	HQ663890
PepMV (EU)	France	*Solanum lycopersicum* L.	NA	AJ438767
	Spain	*Solanum lycopersicum* L.	NA	AJ606360
		*Solanum lycopersicum* L.	NA	NC004067
	Hungry	*Solanum lycopersicum* L.	NA	AM491606
	Belgium	*Solanum lycopersicum* L.	NA	FJ457098
	United Kingdom	*Solanum lycopersicum* L.	NA	KJ018164
	Netherlands	*Solanum lycopersicum* L.	NA	FJ940223
	Poland	*Solanum lycopersicum* L.	NA	JN133846
	United States	*Solanum lycopersicum* L.	NA	JQ314457
			NA	JQ314459
			NA	MN395046
PepMV (CH2)	Belgium	*Solanum lycopersicum* L.	NA	FJ457097
	Italy	*Solanum lycopersicum* L.	NA	HQ663890
	Switzerland	*Solanum lycopersicum* L.	NA	MF422611
	United States	*Solanum lycopersicum* L.	NA	JX866667
	United States	*Solanum lycopersicum* L.	NA	JX866668
	Canada	*Solanum lycopersicum* L.	NA	JX866665
	Canada	*Solanum lycopersicum* L.	NA	JX866666
PepMV (CH1)	Chile	*Solanum lycopersicum* L.	NA	DQ000984
PepMV (US1)	United States	*Solanum lycopersicum* L.	NA	JQ314458
	United States	*Solanum lycopersicum* L.	NA	KF734961
	United States	*Solanum lycopersicum* L.	NA	AY509926
PepMV (US2)	United States	*Solanum lycopersicum* L.	NA	AY509927
ToBRFV	Canada	*Solanum lycopersicum* L.	22NS5 *	OP971902
	Canada	*Solanum lycopersicum* L.	22NS6 *	OP971903
	United States	*Solanum lycopersicum* L.	22NTT13 *	OP971904
	Mexico	*Solanum lycopersicum* L.	22IL16 *	OP971896
	Mexico	*Solanum lycopersicum* L.	22IL17 *	OP971897
	Mexico	*Solanum lycopersicum* L.	22IS19 *	OP971898
	Mexico	*Solanum lycopersicum* L.	22IS20 *	OP971899
	Canada	*Solanum lycopersicum* L.	22AS21 *	OP971890
	Canada	*Solanum lycopersicum* L.	22AS22 *	OP971891
	Mexico	*Solanum lycopersicum* L.	22AL23 *	OP971888
	Mexico	*Solanum lycopersicum* L.	22AL24 *	OP971889
	Mexico	*Solanum lycopersicum* L.	22HNS25 *	OP971894
	Mexico	*Solanum lycopersicum* L.	22HNS26 *	OP971895
	Canada	*Solanum lycopersicum* L.	22HMF28 *	OP971892
	Canada	*Solanum lycopersicum* L.	22HMF29 *	OP971893
	United States	*Solanum lycopersicum* L.	20IG1 *	OP971881
	United States	*Solanum lycopersicum* L.	20NO1 *	OP971882
	United States	*Solanum lycopersicum* L.	20NO2 *	OP971883
	United States	*Solanum lycopersicum* L.	20NVS1 *	OP971900
	United States	*Solanum lycopersicum* L.	20NVS2 *	OP971901
	United States	*Solanum lycopersicum* L.	20NTTG *	OP971884
	United States	*Solanum lycopersicum* L.	20NTTV *	OP971885
	United States	*Solanum lycopersicum* L.	20NVM3 *	OP971886
	United States	*Solanum lycopersicum* L.	20NVM4 *	OP971887
	Jordan	*Solanum lycopersicum* L.	NA	KT383474
	Israel	*Solanum lycopersicum* L.	NA	KX619418
	Israel	*Solanum lycopersicum* L.	NA	OM515237
	Egypt	*Solanum lycopersicum* L.	NA	MN882030
	Turkey	*Solanum lycopersicum* L.	NA	MT107885
	Greece	*Solanum lycopersicum* L.	NA	MN815773
	Italy	*Solanum lycopersicum* L.	NA	MN167466
	France	*Solanum lycopersicum* L.	NA	MW284987
	Germany	*Solanum lycopersicum* L.	NA	MK133095
	Switzerland	*Solanum lycopersicum* L.	NA	OM305070
	Netherlands	*Solanum lycopersicum* L.	NA	OM515245
	United Kingdom	*Solanum lycopersicum* L.	NA	MN182533
	China	*Solanum lycopersicum* L.	NA	MT018320
	Canada	*Solanum lycopersicum* L.	NA	MN549394
	Canada	*Solanum lycopersicum* L.	NA	MN549395
	Canada	*Solanum lycopersicum* L.	NA	MN549396
	Mexico	*Capsicum annuum* L.	NA	MW349655

* Sequences of samples collected in this study.

**Table 3 viruses-15-02305-t003:** Summary of the tomato fruit survey results with the confirmed presence of ToBRFV and PepMV in fruits sold in South Florida groceries in the USA.

	Virus Positive/Total Samples		
Collection Location	PepMV	ToBRFV	Co-Infection
Arcadia	4/4	4/4	4/4
Hollywood	5/6	6/6	5/6
Immokalee	9/11	9/11	7/11
Naples	29/31	26/31	24/31
Total	47/52	45/52	40/52

## Data Availability

All data are presented in the manuscript.

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
