# Peer review of "Co-Infection of Tomato Brown Rugose Fruit Virus and Pepino Mosaic Virus in Grocery Tomatoes in South Florida: Prevalence and Genomic Diversity"

_viruses, 2023, doi:10.3390/v15122305_

Round 1
Reviewer 1 Report
Comments and Suggestions for Authors
In this study, the authors collected tomato fruits from several grocery stores showing necrotic and chlorotic symtoms and tested for PepMV and ToBRFV infections. This then revealed that 86.5% of the selected fruits were infected with ToBRFV while 90% were infected with PepMV alone. However, 73% of the tested fruits had mixed infection. It is an important study showing that plant health inspections carried out at borders may not be sufficient and as shown in this study escapes could occur leading to potential spread of the virus in a country.
The manuscript is written very well and there is no obvious error.
My main concern is the discrepancy in Table 2. For example, in PepMV (US2) row, the origin for samples 20NO1 and 20NO2 was written as Canada, but the origin of these samples in ToBRFV row was written as United States. Could the authors go through the table and correct this and other discrepancies if present.
P1 L35 Please make sure gene nemes are written in italic
P3 L46 Please make Solanum muricatum L. italic
Author Response
We have addressed all comments that were provided by the reviewers in the revised manuscript. All comments were editorial and constructive in nature, which we are grateful for, which improved our manuscript. Therefore, we express our appreciation to anonymous reviewers for their valuable time reviewing our manuscript thoroughly and for suggestions.
The parts of the manuscript modified per the reviewers’ suggestions are highlighted in the revised manuscript. These modifications can also be seen in track changes.
We would happily provide further clarification or additional information if needed.
Sincerely,
Ozgur Batuman
Reviewer 2 Report
Comments and Suggestions for Authors
In the manuscript the authors report the results of an investigation on virus infections in tomato produce from grocery stores in South Florida, US, with focus on Pepino mosaic virus and Tomato brown rugose fruit virus. Following detection, phylogenetic analyses were performed to investigate genetic diversity, and relationships. The results evidenced a significant presence of both viruses, also in mixed infections, both in tomatoes of known origin (Mexico, Canada) or in produce packaged in the US. The genetic diversity studies showed presence of 2 PepMV strains, US2 and CH2; while all TBRFV sequences were grouped in the previously determined Group3. Finally, the authors provide some scenarios that could be associated to a potential spread of the viruses to tomato productions in Florida, where ToBFRV infection in field surveys have not yet been reported. The study is well conducted and provides information on the prevalence of the viruses in tomato produce. The correlation of this to effective pathways of potential distribution and dynamics of further infection in tomato production areas in Florida, as well-remarked by the authors, remains subject of future analyses, and could actually be conducted when infections may indeed be reported. My only remark is that the manuscript describes a snap-shot situation occurring in South Florida, while the generated sequences could be used to investigate virus spread, in particular concerning the PepMV strains and shifts in populations overtime. This would make the manuscript much more interesting and of higher value for the Journal.
My comments on the current version are mostly limited to some editorial suggestions in abstract and introduction.
L12: pathogen of significant importance to
L17: by surveying
L20: were mixed infected
L20-22: this sentence can be moved before the previous one
L24: delete “two”
L31: and cause significant economic losses
L25: “all other R genes”, this should be specific better, there are other R genes not related to Tm
L37: able to overcome
L42: demonstrated, resulting in…
L47: The genome of… consists of a single…
L54: affected by many factors, including
L68: the order of results/discussion sections on ToBRFV and PepMV (or viceversa) can be harmonized throughout the paper
Comments on the Quality of English Language
minor edits necessary
Author Response

(The authors gave the same response as above.)
